# Genes Modulating Butyrate Metabolism for Assessing Clinical Prognosis and Responses to Systematic Therapies in Hepatocellular Carcinoma

**DOI:** 10.3390/biom13010052

**Published:** 2022-12-27

**Authors:** Zhao Chuanbing, Zhang Zhengle, Ding Ruili, Zhu Kongfan, Tao Jing

**Affiliations:** 1Department of Pancreatic Surgery, Renmin Hospital of Wuhan University, Wuhan 430061, China; 2Department of Anesthesiology, Renmin Hospital of Wuhan University, Wuhan 430061, China

**Keywords:** hepatocellular carcinoma, butyrate metabolism, prognosis, immunotherapy, drug sensitivity, TACE

## Abstract

Butyrate, one of the major products of the gut microbiota, has played notable roles in diverse therapies for multiple tumors. Our study aimed to determine the roles of genes that modulate butyrate metabolism (BM) in predicting the clinical prognosis and responses to systemic therapies in hepatocellular carcinoma (HCC). The genes modulating BM were available from the GeneCard database, and gene expression and clinical information were obtained from TCGA-LIHC, GEO, ICGC-JP, and CCLE databases. Candidate genes from these genes that regulate BM were then identified by univariate Cox analysis. According to candidate genes, the patients in TCGA were grouped into distinct subtypes. Moreover, BM- related gene signature (BMGs) was created via the LASSO Cox algorithm. The roles of BMGs in identifying high-risk patients of HCC, assessing the prognoses, and predicting systematic therapies were determined in various datasets. The statistical analyses were fulfilled with R 4.1.3, GraphPad Prism 8.0 and Perl 5.30.0.1 software. In the TCGA cohort, most butyrate-related genes were over-expressed in the B cluster, and patients in the B cluster showed worse prognoses. BMGs constructed by LASSO were composed of eight genes. BMGs exhibited a strong performance in evaluating the prognoses of HCC patients in various datasets, which may be superior to 33 published biomarkers. Furthermore, BMGs may contribute to the early surveillance of HCC, and BMGs could play active roles in assessing the effectiveness of immunotherapy, TACE, ablation therapy, and chemotherapeutic drugs for HCC. BMGs may be served as novel promising biomarkers for early identifying high-risk groups of HCC, as well as assessing prognoses, drug sensitivity, and the responses to immunotherapy, TACE, and ablation therapy in patients with HCC.

## 1. Introduction

Hepatocellular carcinoma has become one of the most prevalent malignancies worldwide, with an incidence rate of 4.7% (seventh of all carcinomas) and a mortality rate of 8.3% (second of all carcinomas) [1]. Despite the great progress of novel therapies, the prognoses of patients with HCC remain unsatisfactory due to its high recurrence and metastasis rates [2,3,4,5]. Therefore, it is of great significance to determine a novel biomarker to predict prognoses and responses to systemic therapies in patients with HCC.

Recently, the gut microbiota was recognized as playing an important role in the potential treatment of various cancers [6]. Research efforts have focused on using metabolites (short-chain fatty acids) of the gut microbiota to exploit cancer therapies. Butyrate, one of the main components of short-chain fatty acids (SCFA), recently received particular attention for its tumor suppressive function and the overall improved outcomes of HCC patients after cancer immunotherapy [7,8]. Indeed, butyrate supplementation significantly improved the outcome of anti-PD-1/PD-L1 immune checkpoint blocker (ICB) therapy in solid cancers [9,10]. Butyrate was indicated to enhance the antitumor viability of T lymphocytes from chimeric antigen receptor (CAR) and cytotoxic T lymphocytes (CTLs) [11,12]. Additionally, butyrate might improve the efficacy of gemcitabine in pancreatic cancer cell lines, the main mechanism of which is the induction of apoptosis in tumor cells [13]. Moverover, the effectiveness and safety of radiation therapy for treating tumors were suggested as being affected by butyrate [14].

Butyrate exerts a similar antitumor effect in HCC. Butyrate was shown to reinforce the anti-HCC effect of sorafenib in vitro and in vivo [15]. Butyrate induces the stagnation of the HCC cell division cycle in the G2/M stage and inhibits the proliferation and invasion of HCC cells [16,17]. Furthermore, butyrate could induce autophagy in HCC cells through the regulatory pathway of reactive oxygen species (ROS) to modulate the development of HCC [18]. It is worth mentioning that butyrate is considered as a potential candidate drug for treating HCC. We hypothesize that it may be more meaningful to investigate the role of butyrate in the progression and treatment of HCC if we focus on genes involved in the regulation of BM. Nevertheless, the functions of genes modulating BM in HCC remain unclear.

Our study aims to identify BM-related subtype and explore the roles of the subtype in predicting the prognoses and responses to immunotherapy and multiple drugs in HCC. According to BM-related genes, we further construct the BMGs. The roles of BMGs in evaluating prognoses and responses to systematic therapies in patients with HCC were determined in various datasets.

## 2. Materials and Methods

### 2.1. Data Acquisition

The set of genes related to butyrate metabolism (Score ≥ 5) was derived from the GeneCards database (http://www.genecards.org, accessed on 18 July 2022). RNA sequencing data and clinical information were available from TCGA-LIHC, ICGC-JP, and GEO (GSE14520, GSE6764, GSE9843, GSE25097, GSE49541, GSE89377, GSE143004, GSE109211, GSE87630, GSE104580, GSE91061, and GSE115821) databases. The gene expression data of HCC cell lines were obtained from the Cancer Cell Line Encyclopedia (CCLE) (https://portals.broadinstitute.org/ccle, accessed on 20 July 2022). Gene mutation information from HCC patients was taken from Genomic Data Commons (GDC) and processed using Perl-5.30.0.1 software (version 5.30.0.1, created by Larry Wall, in Duncan, BC, Canada). The cancer-associated transcription factors were extracted from the Cistrome database http://www.cistrome.org/ (accessed on 10 November 2022). The expression matrix data retrieved from the GEO database were processed by the “limma” and “sva” packages to reduce the confounding bias of these data. In our study, the TCGA-LIHC cohort is the training set cohort, while the others are the validation set cohorts. The main flow of this study is illustrated in Figure 1.

### 2.2. Identification of Butyrate-Related Subtype in HCC

First, we applied the “limma” package to filter differentially expressed genes (DEGs) between HCC tissues and normal liver tissues (|logFc| ≥ 2, Padj < 0.05) among these BM-related genes. Moreover, univariate Cox regression analysis was employed to select candidate genes from the DEGs by the “survival” package (|HR| > 1, FDR < 0.05). Additionally, we performed the correlation network of these candidate genes with “igraph,” “psych,” “reshape2”, and “RColorBrewer” packages.

Furthermore, we have further evaluated the effect of candidate butyrate-related genes on the prognoses and responses to HCC immunotherapy. Based on the expression levels of candidate genes, the “ConsensusClusterPlus” package based on the “Pam” method was employed to classify HCC patients into distinct subtypes.

### 2.3. Prognostic Profiles and Immune Characteristics in Different Subtype Related to BM

Then, the Kaplan-Meier method (KM) was applied to compare overall survival (OS), disease-free interval (DFI), disease-specific survival (DSS), and progression-free interval (PFI) of these distinct clusters with the “survival” package to identify the varied prognoses of these clusters. The univariate and multivariate Cox regression analyses were employed to identify if the BM-related subtype was an independent risk factor of prognoses of HCC patients. Moreover, we determined whether the expression of these genes and the distribution of TNM-stage, vascular invasion, and other indicators differed among these subtypes with the “complexheatmap” package based on the Wilcoxon rank sum test. To further demonstrate the value of the BM-related subtypes in predicting the prognosis of HCC, we perform univariate Cox regression analysis and decision curve analysis (DCA) to compare BM-related subtypes with iClusters [19] and Hoshida [20] subtypes by “survival”, “rms”, and “ggDCA” packages.

According to the mRNA expression of TCGA-LIHC, we used the “immunedeconv” package integrated with six of the latest algorithms (EPIC, TIMER, CIBERSORT, xCELL, MCP-counter, and quanTIseq) to assess the abundance of immune features for distinct subtypes of patients in TCGA-LIHC cohort. The TIDE database was then applied to evaluate immunological therapy responses in patients with HCC [21]. According to the TIDE algorithm, we calculated the TIDE scores, estimated the abundance of cancer-associated fibroblasts (CAF) for each patient in the TCGA cohort, and compared the scores of patients with distinct subtypes in the TCGA cohort using the “ggpubr” package based on the Wilcoxon rank sum test.

Additionally, mRNAsi is an indicator that describes the level of similarity between tumor cells and stem cells. The higher value of mRNAsi indicates a greater degree of aggressiveness of tumor cells. To evaluate aggressiveness of tumor cells, based on the mRNA expression data of TGCA-LIHC cohort, we apply the one-class logistic regression (OCLR) algorithm to estimate the mRANsi for each sample in the TCGA-LIHC cohort [22].

With the application of the “pRRophetic” package, using data from the “Genomics of Drug Sensitivity in Cancer” (GDSC) database, drug sensitivity was determined in patients with varied subtypes of HCC.

### 2.4. The Role of BMGs in Predicting Clinical Prognoses in HCC

#### 2.4.1. Construction and Validation of BMGs

To further reveal the potential roles of the butyric acid metabolic pathway in HCC, the LASSO Cox algorithm was employed to select these optimal prognostic biomarkers from these candidate genes with the application of the package “glmnet”. Ten cross-validations obtained the minimum lambda value, and the coefficients of each variable under this minimum lamba value were exported. Notably, these genes with nonzero coefficients were considered as the most suitable genes comprising BMGs. In this study, the BMGs were constructed as the formula:

BMGs risk score = expression level of gene_A_ ∗ variable coefficient of gene_A_ + expression level of gene_B_ ∗ variable coefficient of gene_B_ + expression level of gene_c_ ∗ variable coefficient of gene_c_ + expression level of gene_i_ ∗ variable coefficient of gene_i_.

In addition, we further verified whether the genes comprising BMGs were differentially expressed in HCC and normal tissues in the ICGC-JP, GSE25097, and GSE87630 cohort with the “ggpubr” package based on the Wilcoxon rank sum test and “heatmap” package. We also identified the expression levels of these genes in HCC cell lines with GraphPad Prism 8 software (version:8.0, created by Harvey Motulsky).

Based on the median risk score, patients were assigned to low- and high-risk groups. The predictive performances of BMGs were assessed by the following analyses. Time-dependent receiver operating characteristic (time-ROC) and concordance index (c-index) analyses were performed by “pec,” “survminer,” “dply,” and “timeROC” packages, and the area under curve (AUC) values of time-ROC and ROC were employed to evaluate the accuracy of BMGs and clinical indicators in assessing prognoses of HCC. Principal component analysis (PCA) was conducted to demonstrate the distribution among patients in these subgroups with the “scatterplot3d” package. The KM method was applied to explore the differences in survival times for patients in these subgroups by “survival” package. Consequently, we applied multivariate Cox regression analysis with the “survival” package to select independent risk factors for prognoses from risk scores and other clinical characteristics. BMGs were externally validated in the ICGC-JP and GSE14520 cohort.

#### 2.4.2. The Correlation between BMGs and Clinical Indicators

To test the underlying roles of BMGs in predicting prognoses of HCC, the correlations between risk score and clinical features, such as TNM-stage, CLIP-stage, BCLC-stage, and vascular invasion, were explored in TCGA-LIHC, ICGC-JP, GSE9843, and GSE14520 cohorts with “complexHeatmap,” “ggpubr,” “limma,” and “reshape2” packages based on the Wilcoxon rank sum test.

#### 2.4.3. Comparison of BMGs with Other Studies

We searched for 33 published studies on HCC gene signatures in the PubMed database, including lactate metabolism, amino acid metabolism, pyroptosis, inflammatory, autophagy, and cuproptosis-related gene signatures [23,24,25,26,27,28,29,30,31,32,33,34,35,36,37,38,39,40,41,42,43,44,45,46,47,48,49,50,51,52,53,54,55]. The exclusive risk scores for these gene signatures were then calculated based on the formulae published in these 33 studies in the TCGA-LIHC cohort. To further determine the predictive performances of BMGs, the c-index analyses were performed to verify the accuracy of BMGs in predicting prognoses of HCC by “survival,” “survcomp,” “ggpubr,” and “ggplot2” packages. Moreover, we performed DCA to identify the clinical application of BMGs in predicting the prognoses of HCC by comparing them with the use of “ggDCA” and “rms” packages.

#### 2.4.4. Construction of a Nomogram Based on BMGs

The nomogram could visualize the results of Cox regression analysis, which has been widely adopted to explore the clinical value of predictive models. To better apply BMGs into clinical practice, we constructed a nomogram based on risk scores using the “rms” and “regplot” packages. The “rms” package was also applied to plot the calibration curve to illustrate the agreement between the predicted 1-, 3-, and 5-year endpoint events and the actual results. Moreover, c-index and time-ROC analyses were performed with the “pec,” “survminer”, and “dplyr” packages to assess the determination of the nomogram and other factors in predicting prognoses of HCC. DCA was performed with the “ggDCA” and “rms” packages to identify the clinical application of the nomogram and risk scores. Moreover, the total score of each patient was calculated based on the nomogram model. By determination of the total score of each patient, the patients were assigned to the high-risk (Nomogram) and low-risk (Nomogram) groups. We applied the KM method to identify whether there were any differences in survival time and survival rate between patients in high-risk (Nomogram) and low-risk (Nomogram) groups. PCA was performed with the “scatterplot3d” package to determine whether the distribution of patients in the subgroups of the nomogram would appear to be clustered.

### 2.5. The Power of BMGs for the Early Detection of the Risk of HCC

Many patients might gradually progress from nonalcoholic fatty liver disease (NAFLD), cirrhosis, and high-grade intraepithelial neoplasia to HCC. However, currently, there is a lack of effective biomarkers that accurately predict the future course of these diseases or distinguish them earlier from HCC and other liver diseases. In our study, we determined whether BMGs could serve as novel biomarkers for the earlier detection of HCC in the GSE6764, GSE49541, and GSE89377 cohorts.

In the GSE6764 cohort, we identified if the risk scores varied in the cirrhosis and HCC groups using “ggpubr” packages based on the Wilcoxon rank sum test. In the GSE89377 and GSE49541 cohorts, we used the same approach to verify whether the risk scores varied among diverse subgroups. ROC curves were performed by the “pROC” package to test the efficacy of risk scores in differentiating between these subgroups as well.

### 2.6. The Significance of BMGs in Assessing the Responses to Diverse Therapies

#### 2.6.1. The Role of BMGs in the Prediction of Responses to Immunotherapy

To evaluate the components of the immune microenvironment in patients with HCC, the ESTIMATE algorithm was employed to explore the tumor microenvironment (TME), including the “ESTIMATE score,” “Stromal score,” and “Immune score” based on the mRNA expression of TCGA-LIHC.

According to mRNA expression of TCGA-LIHC, single sample gene set enrichment analysis (ssGSEA) was performed to estimate the abundance of immune cells as well as immune functions with the “GSVA” package, and the reference gene set of 29 immune-related pathways for this analysis was derived from previous studies [56,57]. The “CIBERSOER” algorithm was applied to evaluate the immune cell infiltration of HCC patients in the TCGA-LIHC cohort, which was run on the basis of gene expression data of immune cells from the “CIBERSORT R script” and LM22 signature. We further identified any differences in TME, immune cells, and immune functions by the “ggpubr” package based on the Wilcoxon test. Additionally, we determined the expression levels of genes related to immunosuppression in two subgroups using the same method.

Given the pivotal role of immune checkpoint expression in assessing the efficacy of immunotherapy in patients with malignancies, we further tested whether the expression levels of immune checkpoints differed between two subgroups with packages “hemtmap,” “ggpubr”, and “ggplot2” based on the Wilcoxon rank sum test. To evaluate the responses to immunotherapy on patients in the two subgroups, the mRNA expression data of TCGA-LIHC were imported into the TIDE [21] and The Cancer Immunome Atlas (TCIA) [58] databases to calculate the TIDE score and immunophenoscore (IPS), respectively. Emerging evidence demonstrated that the T cell inflammation score (TIS), IFN-γ gene signature (IFNG), interferon and antigen presentation (IFNAP), CD8A, SEPEN1, and STAT1 expression might be applied to determine the effect of immunotherapy [59,60,61,62]. Thus, we verified if TIS, IFNG, IFNAP, CD8A, SEPEN1, and STAT1 varied among the TCGA-LIHC cohort in these groups with the “ggpubr” package based on the Wilcoxon rank sum test. We further demonstrated the feasibility of BMGs to predict responses to immunotherapy using data from the GSE91061 and GSE115821 cohorts.

#### 2.6.2. The Role of BMGs in Assessing the Efficacy of TACE and Ablation Therapy

TACE and ablation therapy were considered promising options for advanced HCC patients. Consequently, the clinical significance of investigating the role of BMGs in evaluating responses to TACE and ablation therapy cannot be overstated. We estimated risk scores for each patient in GSE104580 and GSE143004 cohorts, respectively. Furthermore, we have evaluated the performance of BMGs in predicting the responses to TACE and ablation therapy with the “ggpubr” and “pROC” packages.

#### 2.6.3. The Role of BMGs in Assessing the Drug Sensitivity

The “pRRophetic” package was applied to detect effective drugs from more than 300 types of agents for patients in two subgroups, respectively (*p* < 0.001). The correlation analyses were performed between the risk score and the sensitive drugs to obtain the most suitable drugs for patients in these subgroups.

In addition, we determined which group of patients with HCC was more sensitive to “sorafenib”. We assessed the efficacy of BMGs in predicting the sensitivity to “sorafenib” on HCC patients with the “ggbupr” package based on the Wilcoxon rank sum test and “pROC” packages in the GSE109211 cohort, respectively.

Furthermore, drugs that interact with these genes that make up BMGs were derived from the Drug-Gene Interaction Database (DGIdb), and the drug-gene interaction network was visualized using Cytoscape software (V 3.8.0) (created by Lee Hood in Washington, USA).

### 2.7. Other Analyses

#### 2.7.1. Somatic Mutation Analyses

The top 20 mutated genes in these subgroups were exhibited using the “maftool” package. The tumor mutation burden (TMB) of each sample was calculated by the “maftool” package. Then, we identified whether there were any differences in TMB between these subgroups with the “ggpubr” package based on the Wilcoxon rank sum test.

#### 2.7.2. Analysis of Molecular Functions and Pathways

To explore the distribution of tumor-related pathways in these subgroups, we collected genes involved in tumor-related pathways and calculated the enrichment scores for each sample in each pathway according to the ssGSEA algorithm using “GSVA” package. Additionally, we further determined if there existed any distinctions in these pathways between high- and low-risk groups with the “ggpubr” package based on the Wilcoxon rank sum test.

Furthermore, to explore the distinct molecular functions of patients in the high- and low-risk groups, the “limma” package was used to analyze DEGs in the two subgroups (|logFC|≥ 1, FDR < 0.05). Then, we conducted the Gene Ontology (GO) and Kyoto Encyclopedia of Genes and Genomes (KEGG) analyses of these DEGs using the “clusterProfiler” package. To explore the differences in the potential pathways between these subgroups, the gene set enrichment analysis (GSEA) was conducted between the high and low-risk groups (*p* < 0.05) with reference gene sets from the MsigDB database.

Given the significance of genes constituting BMGs in predicting prognoses and systemic therapies of HCC, we further uncovered the transcription factors of these genes. Consequently, we performed a correlation analysis of cancer-associated transcription factors with these genes to obtain the particular transcription factors of these genes using the “corrplot” package (|r > 0.35|, FDR < 0.05). The network of these genes and particular transcription factors was constructed by Cystocape (V 3.8.0) software (created by Lee Hood in Washington, USA).

#### 2.7.3. Stemness Indices Analyses

To verify the aggressiveness of tumor cells in the high- and low-risk groups of patients, the “ggpubr” package based on the wilcoxon rank sum test was then employed to assess the distinctions in mRNAsi between the two subgroups.

### 2.8. Statistical Analyses

The statistical analyses of this study were fulfilled with R 4.1.3, GraphPad Prism 8.0, and Perl 5.30.0.1 software.

## 3. Results

### 3.1. Identification of Candidate Genes

We derived a total of 757 genes modulating metabolism in the GeneCards database (Score ≥ 5). Figure 2A illustrated that 75 DEGs from these 757 genes were filtered out. Consequently, univariate Cox regression analysis was employed to screen 41 candidate genes from these 75 genes and the candidate genes significantly interacted (Figure 2B).

### 3.2. Identification of Butyrate Metabolism-Related Subtype in HCC

The HCC samples in TCGA-LIHC were classified into A and B clusters, according to the expression of these candidate genes (Figure 2C,D). Figure 2E presented that patients were clearly clustered in the A and B subgroups. The worse prognoses were observed in patients of the B subtype, and the statuses with vascular invasion, advanced TNM-stage and grade might be more commonly observed in B subtypes (Figure 2F–J). The enhanced expression levels of most candidate genes were markedly enriched in subtype B (Figure 2J). Furthermore, we discovered that butyrate metabolism-related subtype was an independent risk factor that impacted the prognosis of HCC (Figure 2K,L). As shown in Appendix A, BM-related subtype had better clinical values than iCluster and Hoshida cluster in predicting the prognosis of HCC.

As shown in Figure 3A,B, immune cell infiltration was significantly distinct in A and B subtypes. Higher expression levels of immune checkpoints were enriched in subtypes (Figure 3C). These findings demonstrated that patients in the B cluster presented a significant state of immunosuppression compared to patients in the A cluster.

As for mRNAsi, it was higher in cluster B than in cluster A, which suggested that tumor cells may be more aggressive in patients with type B (Figure 4C). Compared to cluster A, the CAF in cluster B was at greater levels (Figure 4A). The elevated TIDE scores were enriched in the A cluster (Figure 4B). The IC50 values for axitinib, AICAR, and AMG.706 were higher in cluster B, while the IC50 values for AUY922, ATRA, and AG.014699 were higher in cluster A (Figure 4D–I). These results implied that cluster B exhibited a better response to immunotherapy, while cluster A group had greater effects on axitinib, AICAR, and AMG.706.

### 3.3. Excellent Performance of BMGs to Predict the Prognosis of HCC

#### 3.3.1. Construction and Validation of BMGs in Diverse Cohorts

Considering the potential of butyrate-related subtypes to assess the clinical prognosis and treatment, BMGs were constituted by eight genes (LCAT, G6PD, SPP1, GLP1R, GAD1, MMP1, CCNA2, and MAPT), which were obtained by the LASSO Cox algorithm from 41 candidate genes (Figure 5A,B). According to the ICGC, GSE25097, GSE87630, and CCLE datasets (Figure 5C–F), the G6PD, SPP1, GLP1R, GAD1, MMP1, CCNA2, and MAPT were overexpressed in HCC tissues, while LCAT was lowly expressed in HCC tissues.

BMGs risk score = expression level of G6PD ∗ 0.1502 − expression level of LCAT ∗ 0.0445 + expression level of SPP1 ∗ 0.0235 + expression level of GLP1R ∗ 0.2587 + expression level of GAD1 ∗ 0.1023 + expression level of MMP1 ∗ 0.0600 + expression level of CCNA2 ∗ 0.0333 + expression level of MAPT ∗ 0.1375.

Our results indicated that patients with higher risk scores presented poorer prognoses (Figure 6A,F–I). The PCA demonstrated a significant clustering of the two subgroups (Figure 6B). As shown in Figure 6C, BMGs were the strongest predictors of prognoses in patients with HCC. Additionally, BMGs seemed to be superior in assessing the prognoses of HCC patients compared with the TNM stage, gender, grade, and age (Figure 6D,E). Figure 6K,L displayed that BMGs were the independent risk factors resulting in the worse prognoses. Moreover, we discovered that a strong association between BM subtype, BMGs and survival time in the TCGA-LIHC cohort (Figure 6J).

As shown in Figure 7A–H, the prediction performance of BMGs in the ICGC-JP cohort is generally per that of the TCGA-LIHC cohort. Similarly, we indicated that BMGs showed great power in assessing the prognoses of patients with HCC in the GSE14520 cohort (Appendix A). Collectively, BMGs could be applied as effective biomarkers to evaluate the prognoses of patients with HCC.

#### 3.3.2. The Correlation between BMGs and Clinical Indicators

In the TCGA-LIHC cohort, elevated risk scores were observed in patients with advanced TNM-stages and tumor grades, as well as in patients with tumor invasion of blood vessels (Figure 8A). The results in the ICGC-JP, GSE9843, and GSE14520 cohorts were approximately the same as those of the TCGA cohort (Figure 8B–E). These findings suggested that BMGs performed excellently in predicting prognoses for HCC patients.

#### 3.3.3. Stronger Predictive Performances of BMGs

As shown in Figure 9A,B and Figure 10A,B, the c-index of the BMGs to evaluate the prognoses of patients with HCC was 0.702, which was higher than the 33 promising gene signatures. It was noteworthy that BMGs presented with stronger clinical values in determining prognoses of HCC patients compared with other gene signatures (Figure 9C–K and Figure 10C–K, Appendix A). These results verified the excellent predictive performances of BMGs to evaluate prognoses in HCC patients.

#### 3.3.4. Construction of BMGs-Based Nomogram

To comprehensively apply BMGs to clinical practice, the nomogram based on BMGs, gender, TNM-stage, and age was created in the TCGA-LIHC cohort (Figure 11A). The AUC values of 1- to 3- and 5-year of the nomogram for the prediction of the prognoses of HCC were 0.804, 0.741, and 0.718, respectively (Figure 11B). As shown in Figure 11D, the c-index value of nomogram was higher than the risk score and TNM stage. These results demonstrated the superior accuracy of the nomogram compared with the risk score and TNM stage. DCA showed that the nomogram could achieve better clinical promotion efficacy than BMGs (Figure 11G). Moreover, the calibration curves showed a high agreement between the predicted and actual values of 1-, 3-, and 5-year survival (Figure 11C). According to the median of the total score of nomogram, patients were classified into the high-risk (Nomogram) and low-risk (Nomogram) groups, respectively. In addition, the nomogram could discriminate between high-risk (nomogram) and low-risk (nomogram) patients, the former of which showing worse OS (Figure 11E,F). Our results indicated that the BMGs-based nomogram was worth promoting in clinical practice. Taken together, the nomogram could be applied as a more effective predictor of prognoses for patients with HCC than BMGs.

### 3.4. The Great Potential of BMGs for Determining the High-Risk Group of HCC

Higher risk scores were observed in HCC patients, and BMGs could distinguish between patients with cirrhosis and HCC with the AUC value of 0.892 (Figure 12A,B). Furthermore, we further identified a trend of gradually increasing risk scores in the HCC group compared to patients with intraepithelial neoplasia and chronic hepatitis (Figure 12C). The AUC value of risk score to differentiate patients with HCC and intraepithelial neoplasia was 0.822 (Figure 12D). Since patients with advanced steatohepatitis have a greater probability of developing HCC, we verified if any distinctions in risk scores between patients in the two subgroups existed. The results indicated that the enhanced risk scores were clustered in advanced steatohepatitis, and the AUC value for patients with advanced steatohepatitis assessed by risk scores was 0.792 (Figure 12E,F). These results demonstrated that BMGs might play vital roles in identifying a high-risk group of HCC, which may greatly benefit for the early diagnosis and prevention of HCC.

### 3.5. The Role of BMGs in Predicting Responses to Systematic Therapies

#### 3.5.1. The Excellent Predictive Performances in Predicting Immunotherapy Responses

ESTIMATE analysis showed significant distinctions between the two subgroups in the tumor microenvironment (TME) existed (Figure 13A). ssGSEA analysis and the CIBERSORT algorithm demonstrated an increased abundance of Th2 cells and Treg cells enriched in the high-risk group, while elevated NK cells and monocytes clustered in the low-risk group (Figure 13B,C). In terms of immune function, the APC_co_inhibition, T cell_co inhibition, and immune checkpoint were observed in another group, while type-I IFN and type-II IFN responses seemed stronger in the low-risk group (Figure 13D). The expression level of genes modulating immunosuppression was overexpressed in the high-risk group (Figure 13E). Moreover, we found that these immune checkpoints were highly expressed in the high-risk group (Figure 13F). Our study indicated that patients in the high-risk group were more likely to experience the immune escape of tumors.

Figure 14A,B revealed that patients in the high-risk group presented reduced TIDE scores, implying that patients in the high-risk group might be more likely to gain benefits from immunologic therapy. Figure 14I illustrated that the low-risk patients appear to be insensitive to immunologic therapy. Importantly, enhanced TIS, NFAG, IFNAP, CD8A, STAT1, and SEPEN1 were clustered in the high-risk group (Figure 14C–H). These results suggested that patients in the high-risk group might be more susceptible to immunologic therapy. Notably, in the GSE91061 and GSE115821 cohorts, BMGs could also predict the effect of immunotherapy on malignant tumors (Figure 14J,K). These results illustrated the key roles of BMGs played in evaluating the responses to immunologic therapy of malignant tumors, especially in HCC.

#### 3.5.2. BMGs for Evaluation of Responses to TACE and Ablation Therapy

Our study showed that patients with HCC susceptible to TACE therapy presented reduced risk scores (*p* < 0.01), and the AUC of the ROC curve for the risk score assessment of response to TACE was 0.761 (Figure 15A,B). Moreover, enhanced risk scores were enriched in patients with complete response (CR) after ablation therapy. The AUC value of BMGs for assessing the efficacy of ablation therapy was 0.783 (Figure 15C,D). This means that those in the high-risk group might receive benefits from ablative therapy. We further observed that the risk scores of patients with CR and partial response (PR) were significantly increased in the post-ablation therapy compared to the pre-ablation therapy (Figure 15E). This implied that regulation of the eight genes constituting BMGs might improve the outcome of ablative therapy in patients with HCC.

#### 3.5.3. BMGs for Prediction of Drug Sensitivity

We discovered that patients in the high-risk group may be more sensitive to ‘sorafenib’, and the AUC value to evaluate the response to ‘sorafenib’ of BMGs was 0.895 in the GSE109211 cohort (Figure 15F,G).

We used the “pRRophetic” package to screen for sensitive drugs for patients in the two subgroups in the TCGA cohort, respectively (*p* < 0.001). Our results uncovered that patients in the high-risk group seemed to be more sensitive to “sunitinib”, “rapamycin,” and “PHA-665752”, while patients in another group seemed to be susceptible to “erlotinib” (Figure 15H,I). As shown in Appendix A, drugs interacting with the eight genes that make up the BMGs were obtained from the DGIdb.

### 3.6. Analysis of Molecular Function in Different Groups

We discovered that common tumor-associated pathways were clustered in the high-risk group, such as DNA repair, angiogenesis, apoptosis, ferroptosis, and tumor inflammation (Figure 16A).

GSEA analysis showed that many cancer metastatic pathways were enriched in the high-risk group, while several metabolic pathways were enriched in the other group (Appendix A). GO analysis revealed that there was a significant enrichment of multiple immune-associated pathways (Appendix A). These results suggested that the molecular functions of BMGs may be mainly related to metabolic proliferation and immune regulation.

As shown in Figure 16B, the network of these eight genes and particular transcription factors was constructed. According to this network, we may modulate the expression of these eight genes by regulating transcription factors. These results might provide innovative ideas to find new targets for treatment of HCC.

### 3.7. Stemness Indices Analysis

As shown in Appendix A, mRNAsi were at greater levels in the high-risk group than in the low-risk group. The risk score was positively correlated with mRNAsi (Appendix A). As shown in Appendix A, patients with higher mRNAsi in HCC presented worse prognoses. This may be one of the main reasons for the worse prognoses of patients in the high-risk group.

### 3.8. Somatic Mutation Analyses

As shown in Appendix A, the most common mutated genes varied in the high- and low-risk groups, and the most common mutated genes in the high- and low-risk groups were TP53 and CTNNB1, respectively. In addition, TMB could significantly affect the survival rate of patients with HCC (Appendix A). Of interest, there were no significant differences in TMB between the high- and low-risk groups (Appendix A). However, combining TMB with risk score could be more helpful for the risk stratification in patients with HCC (Appendix A).

## 4. Discussion

Butyrate, one of the major products of the gut microbiota, has been reported to play a significant antitumor role in various tumors, such as colon cancer, breast cancer, and gastric cancer, etc. [63,64,65,66]. Emerging studies suggested that butyrate inhibits the progression of HCC through multiple mechanisms [16,17]. Moreover, butyrate may impact the effectiveness of immunotherapy in solid tumors [7]. These studies demonstrated that butyrate might have great potential as a novel alternative for treating HCC. Nevertheless, the relevance of the butyric acid metabolic pathway to the prognosis and systemic therapies of HCC has not been comprehensively studied. Thus, we aimed to elucidate the role of the butyric acid metabolic pathway in the early diagnosis, prognosis assessment, and systemic treatment of HCC.

In our study, according to the expression levels of candidate BM-related genes, the HCC samples in the TCGA-LIHC cohort were grouped into A and B clusters, with most of the genes highly expressed in the latter. Patients in the B cluster showed worse prognoses and may be more likely to respond to immunotherapy. Our results suggested that BM-related subtype could be applied to predict prognoses and immunotherapy responses in patients with HCC.

To further reveal the underlying prognostic and molecular mechanisms of butyrate metabolism pathways for early diagnosis, prognoses, and systemic treatment, the BMGs were constructed by the LASSO Cox algorithm. The BMGs were comprised of eight genes, including LCAT, G6PD, SPP1, GLP1R, GAD1, MMP1, CCNA2, and MAPT. Recent evidence indicated that G6PD deficiency might inhibit the carcinogenesis, proliferation, and metastasis of HCC cells by upregulating cytochrome P450 oxidoreductase (POR), suggesting that G6PD may be applied as a biomarker for the treatment of HCC in the future [67]. CCNA2 facilitates cell cycle progression in HCC cells and is considered a molecular stimulator of HCC progression [68]. In addition, MMP1-regulated signaling pathways have been indicated to be involved in the metastasis and progression of HCC [69,70]. GAD1 has been reported to stimulate tumor cell invasion and metastasis by regulating β-catenin translocation and activating MMP7 [71]. GLP1R may stimulate the progression of HCC via the cAMP-PKA-EGFR-STAT3 axis [72]. It should be noted that the down-regulation of MAPT expression is a reliable predictive marker of drug sensitivity in tumor cells [73]. In our study, the network of these genes and particular transcription factors was constructed, which may provide innovative ideas for finding new targets for the treatment of HCC. Altogether, our study might establish a groundwork for upcoming studies, but the mechanisms of these genes in HCC still deserve further validation.

BMGs were established and validated in multiple datasets, including the TCGA-LIHC, ICGC-JP, GSE14520, GSE6764, and GSE9843 cohorts. We discovered that BMGs played a remarkable role in assessing the prognoses of HCC. Compared to the 33 published gene signatures in the PubMed database, BMGs may be more advantageous in evaluating the prognoses of HCC patients. Significantly, BMGs could distinguish HCC from other liver diseases during the early course of HCC based on GSE6764, GSE89377, and GSE49451 cohorts. More importantly, BMGs showed excellent predictive performances for evaluating the responses to diverse therapies, including immunotherapy, TACE therapy, ablation therapy, and drug sensitivity based on TCGA-LIHC, TCIA, TIDE, GSE91061, GSE115821, GSE104580, GSE143004, and GSE109211 cohorts. Our study demonstrated that patients in the high-risk group could gain benefits from immunologic and ablation therapy, while these patients in the low-risk group might be sensitive to TACE treatment and susceptible to “sorafenib”. These findings suggested that BMGs played a significant role in assessing the prognoses and multiple therapies of HCC. Furthermore, the most commonly mutated genes and potential molecular functions were distinct between these subgroups.

In the GSE6764, GSE89377, and GSE49541 cohorts, we discovered that BMGs could have the potential to serve the innovative biomarkers to identify a high-risk group of developing HCC, which was beneficial for the early diagnosis of HCC.

BMGs performed excellently in assessing the prognoses of HCC, which might be the independent risk factors for HCC prognoses. Importantly, compared to the other 33 published biomarkers, BMGs have demonstrated a superior potential to predict the prognosis of HCC. The nomogram based on the risk score was constructed, and all calibration, ROC, and c-index curves indicated the great clinical values of the nomogram in assessing the prognoses of HCC.

Based on BMGs, we might speculate on potential factors for varying prognoses between high- and low-risk groups, such as genetic mutation profile, mRNAsi, and immune features.

We hypothesized that genetic mutation profile might contribute to this outcome. Our study showed that a higher total mutation frequency was observed in the high-risk group. Notably, enhanced genomic instability is strongly associated with the prognosis of malignant neoplasms [74]. Somatic mutation analysis demonstrated that the most common genetic mutations in the high- and low-risk groups were TP53 and CTNNB1, respectively. Recent evidence suggests that HCC patients with a higher frequency of CTNNB1 mutations tend to present smaller and better-differentiated tumors [75]. Conversely, the phenomenon of vascular invasion in malignant tumors is closely associated with mutations in TP53 [75]. It is evident that genomic mutation profiles can result in distinct prognoses of high- and low-risk populations.

Profiles of molecular functions may also lead to varied prognoses in patients in two subgroups as well. As shown in Figure 10A, the more abundant pathways involved in tumorigenesis and proliferation were concentrated in patients in the high-risk group. In addition, the GSEA analysis revealed that many invasion and proliferation pathways in cancers were enriched in the high-risk group, while several metabolic pathways clustered in the other group.

Our results indicated significant differences in TME. In regard to immune cells, a greater amount of Th2 and Treg cells and a lower amount of NK cells were detected in the high-risk group. Immune functions also varied between subgroups. Our results suggested that there was a marked state of immunosuppression in the high-risk group, which contributed to tumor invasion and metastasis [76]. Altogether, immune features may also be responsible for the markedly varied prognoses of high- and low-risk groups.

Furthermore, we identified enhanced mRNAsi in the high-risk group of patients. Moreover, the mRNAsi was positively associated with risk scores. It has been reported that higher mRNAsi represented tumor cells with more aggression [77]. Our study suggested that the stemness index profiles were another reason for the varied prognoses of the two subgroups.

According to BMGs in our study, we may implement more tailored treatment modalities for patients with HCC, especially regarding immunotherapy, TACE therapy, ablation therapy, and chemotherapy.

Immunotherapy is an effective therapy for some patients with HCC. However, one of the prerequisites to ensure the efficacy of immunotherapy is the ample expression of immune checkpoints in tumor tissues. In these years, immune checkpoint inhibitors (ICISs) targeting CTLA-4 and PDCD1 have achieved good efficacy for HCC patients [78]. Our results indicated that elevated expression levels of common checkpoints were enriched in the high-risk group. Additionally, TIDE and IPS scores confirmed that patients in the high-risk group could have a greater potential to obtain favorable outcomes from immunological therapy. Our study demonstrated that enhanced TIS, NFAG, and IFNAP in the high-risk group verified that patients in the high-risk group tended to respond to immunotherapy. Moreover, CTNNB1 mutations were more common in the low-risk group, and emerging evidence suggested that patients with higher CTNNB1 were more likely to be resistant to immunotherapy [79]. Our study initially showed that these regulatory genes could predict the efficacy of immunotherapy. Previous studies have indicated that butyrate supplementation could improve the effectiveness of immunotherapy in patients with malignant tumors. In the future, regulating BM related-genes to enhance the efficacy of immunotherapy in HCC patients might be a promising therapeutic direction.

TACE and ablation treatment are considered promising alternatives for patients with advanced HCC. However, in clinical practice, we found that not all patients with advanced HCC responded to TACE and ablation therapy. Our research initially addressed this challenge. Our study suggested that patients in the low-risk group might be more sensitive to TACE, while patients in the other group might respond to ablation therapy. Taken together, BMGs might be the effective biomarkers to assess the effectiveness of TACE and ablation therapy for HCC.

Chemotherapy remains the conventional treatment modality for a large proportion of patients with advanced HCC. However, faced with more than 300 chemotherapeutic agents, selecting the appropriate drug for special HCC patients has always been a puzzling problem for clinicians. Based on BMGs, sensitive chemotherapeutic drugs were screened for two subgroups, respectively. Our results revealed that patients in the high-risk group might have greater chances of gaining benefits from ‘paclitaxel’. Notably, we observed that tumors in the low-risk group of patients might be more vulnerable to “sorafila”. Altogether, based on BMGs, we could develop individualized drug treatment strategies, which may be critical in improving the prognoses of HCC.

Our study focused on the role of genes related to butyrate metabolism in the prognosis and systemic therapies of HCC in various datasets, which was innovative and few were previously reported. It was noteworthy that BMGs performed more favorably in evaluating the OS of HCC patients compared to 33 potential gene signatures in the PubMed database. In addition, our study is the first to systematically elucidate the role of gene signatures in the early detection of HCC, predicting responses to immunotherapy, drug sensitivity, TACE therapy, and ablative therapy in HCC, which was of great clinical promotion. Moreover, 16 datasets from different databases might increase the credibility of the findings. However, since these cohorts in this study were collected on various public platforms, the presence of tumor heterogeneity in patients included in different data sets is inevitable. We demonstrated that BMGs were superior to other gene signatures in predicting HCC prognoses in the TCGA-LIHC data set, without validation in other data sets, which is also a limitation of this study. The role of genes modulating butyrate metabolism in evaluating the prognoses and systemic therapies of HCC remains to be confirmed by prospective cohort studies and basic research in large samples.

## 5. Conclusions

Our study has defined a novel gene signature (BMGs) based on eight genes that modulate butyrate metabolism. Our findings suggested that BMGs performed excellently in the detection of the high-risk group of developing HCC and in predicting prognoses, as well as responses to immunotherapy, TACE, and ablation therapy in patients with HCC. According to BMGs, potentially effective drugs were selected from multiple drugs for patients in the high- and low-risk groups, respectively. Therefore, it can be expected that BMGs will serve as novel promising biomarkers to predict prognoses and responses to systemic therapies. We believe that our results will lay the foundation for future studies on the metabolites of the gut microbiota in HCC.

## Figures and Tables

**Figure 1 biomolecules-13-00052-f001:**
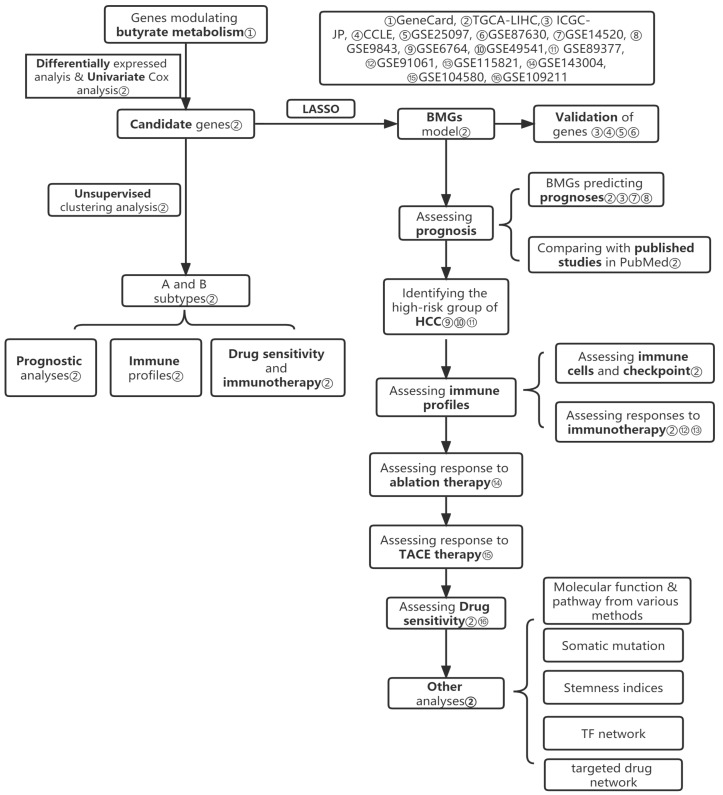
The flowchart of our study.

**Figure 2 biomolecules-13-00052-f002:**
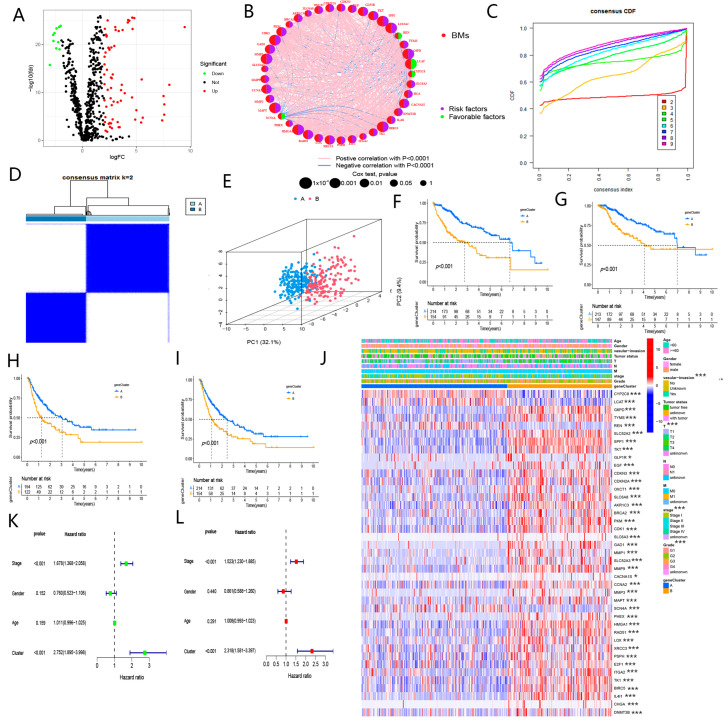
Consensus clustering of butyrate-related genes in HCC. (**A**) Differential expression analysis of butyrate metabolism-related genes in HCC tissues and normal tissues. (**B**) The correlation of candidate genes. (**C**) The consensus CDF value of patients with using unsupervised consensus clustering methods (K-means) in the TCGA-LIHC cohort. (**D**) The consensus matrix of patients with k = 2 using 1000 iterations of unsupervised consensus clustering methods (K-means) to ensure the stability of the clusters in the TCGA-LIHC cohort. (**E**) PCA analysis of distinct candidate butyrate clusters in the TCGA-LIHC cohort. (**F**) KM curves of overall survival (OS) for distinct clusters. (**G**) KM curves of disease-specific survival (DSS) for distinct clusters. (**H**) KM curves of disease-free interval (DFI) for distinct clusters. (**I**) KM curves of progression-free interval (PFI) for distinct clusters. (**J**) Complex heatmap showing the distribution of candidate butyrate genes and clinical features in A and B clusters. (**K**) Univariate Cox analysis of cluster, TNM-stage, age, and gender. (**L**) Multivariate Cox analysis of cluster and TNM-stage. (“*” and “***” represent *p* < 0.05, and *p* < 0.001, respectively).

**Figure 3 biomolecules-13-00052-f003:**
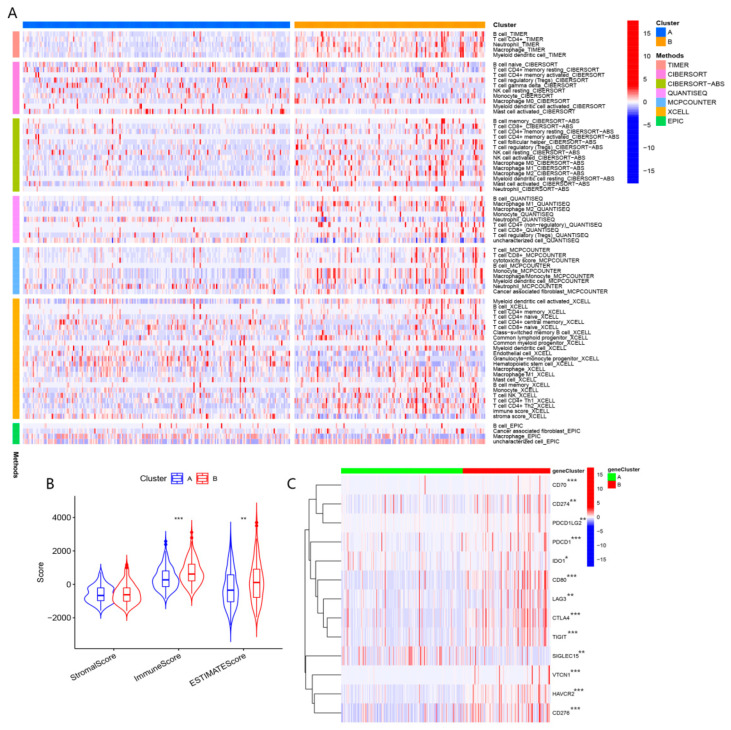
(**A**) Comparison of immune cell infiltration in clusters A and B based on seven algorithm. (**B**) Comparison of TME in clusters A and B. (**C**) Comparison of expression levels of immune checkpoint in clusters A and B. (“*”, “**”, “***” represent *p* < 0.05, *p* < 0.01, *p* < 0.001, respectively).

**Figure 4 biomolecules-13-00052-f004:**
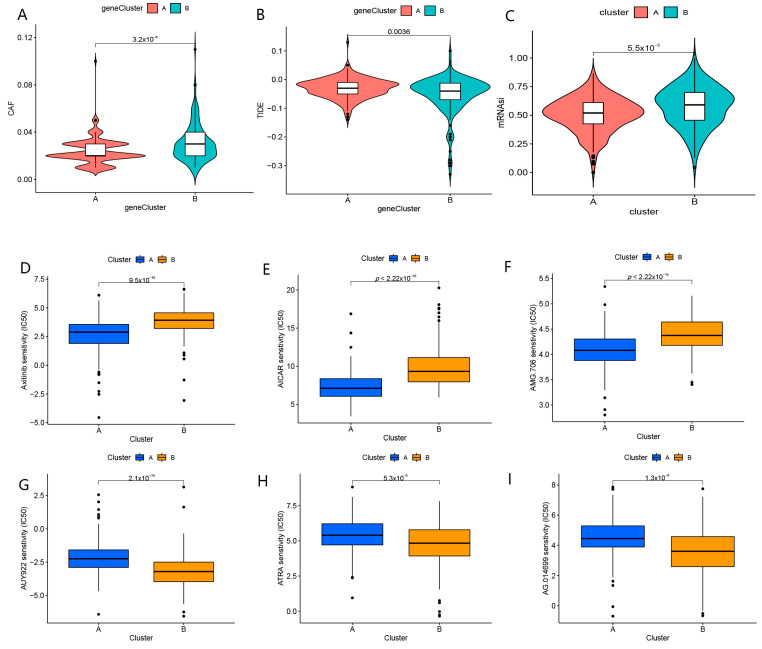
Responses to immunotherapy and drug sensitivity in distinct subtypes. (**A**) Comparison of CAF in clusters A and B. (**B**) Comparison of TIDE scores in clusters A and B. (**C**) Comparison of mRNAsi of immune checkpoint in cluster A and B. (**D**) Comparison of IC50 of Axitinib in clusters A and B. (**E**) Comparison of IC50 of AICAR in cluster A and B. (**F**) Comparison of IC50 of AMG.706 in clusters A and B. (**G**) Comparison of IC50 of AUY922 in clusters A and B. (**H**) Comparison of IC50 of ATRA in clusters A and B. (**I**) Comparison of IC50 of AG.014699 in clusters A and B.

**Figure 5 biomolecules-13-00052-f005:**
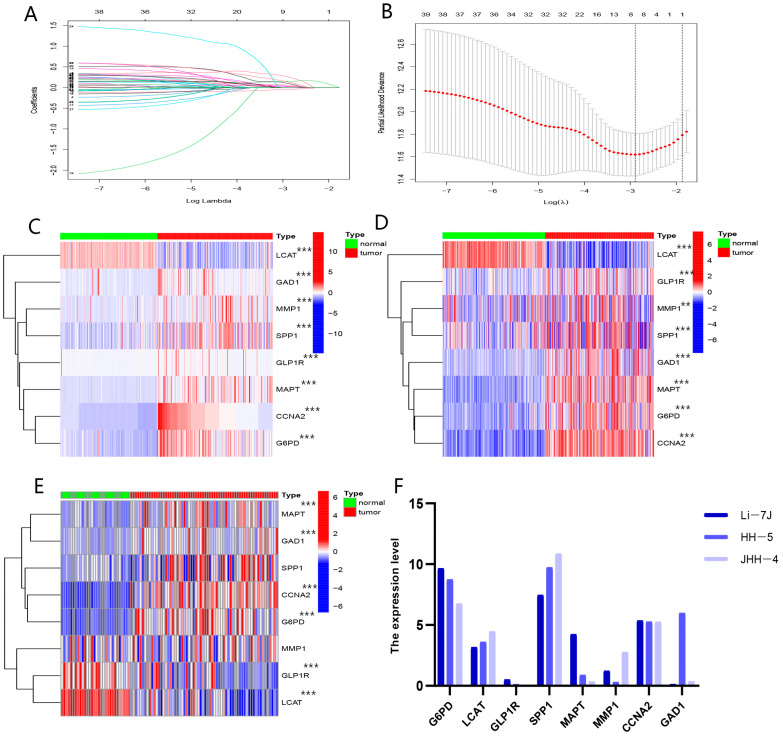
The procession of constructing the BMGs. (**A**) LASSO coefficient profiles. (**B**) Candidate genes were filtered by the LASSO algorithm. (**C**) Validation of the expression of the eight genes that make up the BMGs between HCC and normal tissues in ICGC-JP. (**D**) Validation of the expression of the eight genes that make up the BMGs between HCC and normal tissues in GSE25097. (**E**) Validation of the expression of the eight genes that make up the BMGs between HCC and normal tissues in GSE87630. (**F**) Validation of the expression of the eight genes that make up the BMGs in HCC cell lines in CCLE. (“**”, “***” represent *p* < 0.01, *p* < 0.001, respectively.).

**Figure 6 biomolecules-13-00052-f006:**
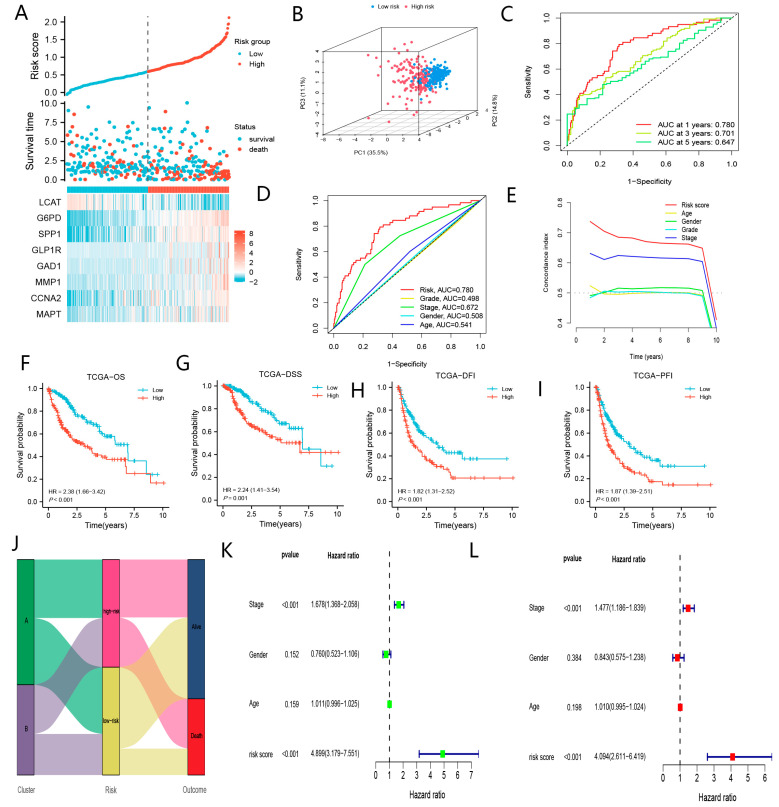
The excellent predictive performances of BMGs in TCGA-LIHC. (**A**) The risk score distribution plot. (**B**) PCA analysis of low- and high-risk groups. (**C**) The timeROC curve of BMGs in predicting the prognoses of HCC. (**D**) The ROC curve of risk score, TNM-stage, age, and gender in predicting the prognoses of HCC. (**E**) The c-index curve of risk score, TNM-stage, age, and gender in predicting the prognoses of HCC. (**F**) KM curve of OS for patients in low- and high-risk groups. (**G**) KM curve of DSS for patients in low- and high-risk groups. (**H**) KM curve of DFI for patients in low- and high-risk groups. (**I**) KM curve of PFI for patients in low- and high-risk groups. (**J**) Sankey plot summarized the correlation among the clusters, BMGs, and survival status. (**K**) Univariate Cox analysis of risk score, TNM-stage, age, and gender. (**L**) Multivariate Cox analysis of risk score and TNM-stage.

**Figure 7 biomolecules-13-00052-f007:**
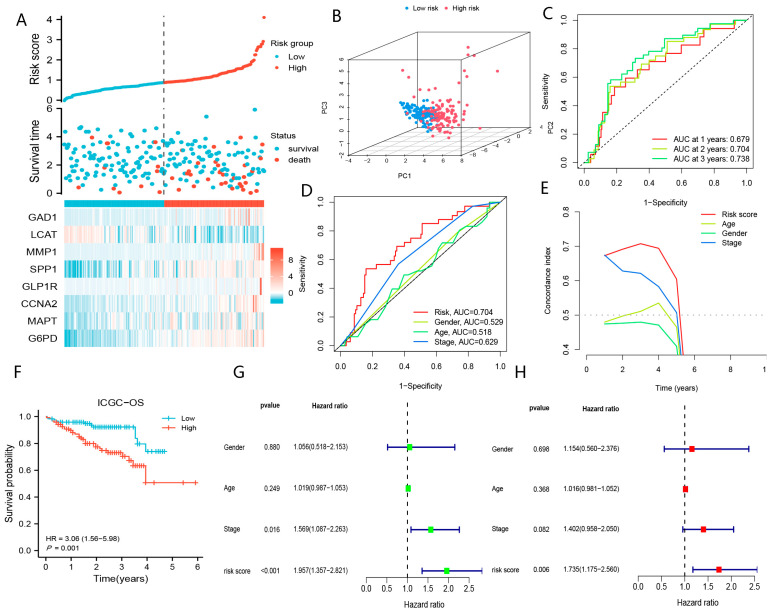
The excellent predictive performances of BMGs in ICGC-JP. (**A**) The risk score distribution plot. (**B**) PCA plot low- and high-risk groups. (**C**) The timeROC curve of BMGs in predicting the prognoses of HCC. (**D**) The ROC curve of risk score, TNM-stage, age, and gender in predicting the prognoses of HCC. (**E**) The c-index curve of risk score, TNM-stage, age, and gender in predicting the prognoses of HCC. (**F**) KM curve of OS for patients in low- and high-risk groups. (**G**) Univariate Cox analysis of risk score, TNM-stage, age, and gender. (**H**) Multivariate Cox analysis of risk score and TNM-stage.

**Figure 8 biomolecules-13-00052-f008:**
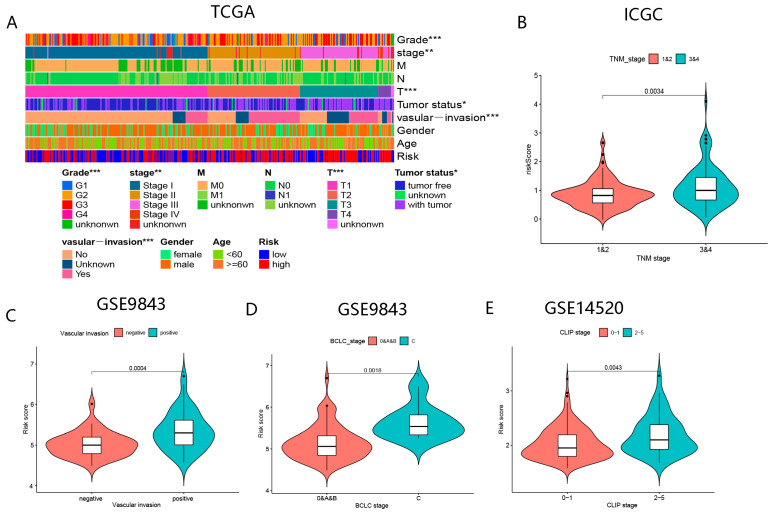
The correlation of risk scores with clinical indicators in diverse cohorts. (**A**) The correlation of risk scores and TMN-stage, vascular invasion, tumor status, and grade in TCGA-LIHC. (**B**) Comparison of risk scores between diverse TNM-stage in ICGC-JP. (**C**) Comparison of risk scores between positive and negative vascular invasion in GSE9843. (**D**) Comparison of risk scores between diverse BCLC-stage in GSE9843. (**E**) Comparison of risk scores between diverse CLIP-stage in GSE14520. (“*”, “**”, “***” represent *p* < 0.05, *p* < 0.01, *p* < 0.001, respectively).

**Figure 9 biomolecules-13-00052-f009:**
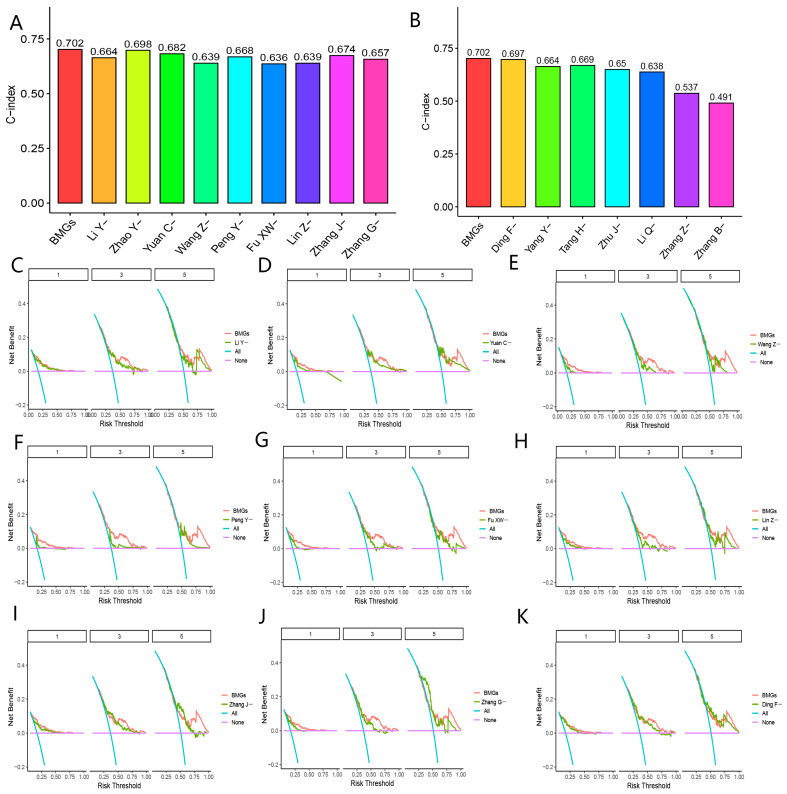
(**A**) BMGs and nine gene signatures in PubMed databases for prediction of the prognoses of HCC. (**B**) C-index analyses about BMGs and seven gene signatures in PubMed databases for prediction of the prognoses of HCC. (**C**) Decision curve analyses of BMGs and the Li Y-study. (**D**) Decision curve analyses of BMGs and the Yuan C-study. (**E**) Decision curve analyses of BMGs and the Wang Z-study. (**F**) Decision curve analyses of BMGs and the Peng Y-study. (**G**) Decision curve analyses of BMGs and the Fu XW-study. (**H**) Decision curve analyses of BMGs and the Lin Z-study. (**I**) Decision curve analyses of BMGs and the Zhang J-study. (**J**) Decision curve analyses of BMGs and the Zhang G-study. (**K**) Decision curve analyses of BMGs and the Ding F-study.

**Figure 10 biomolecules-13-00052-f010:**
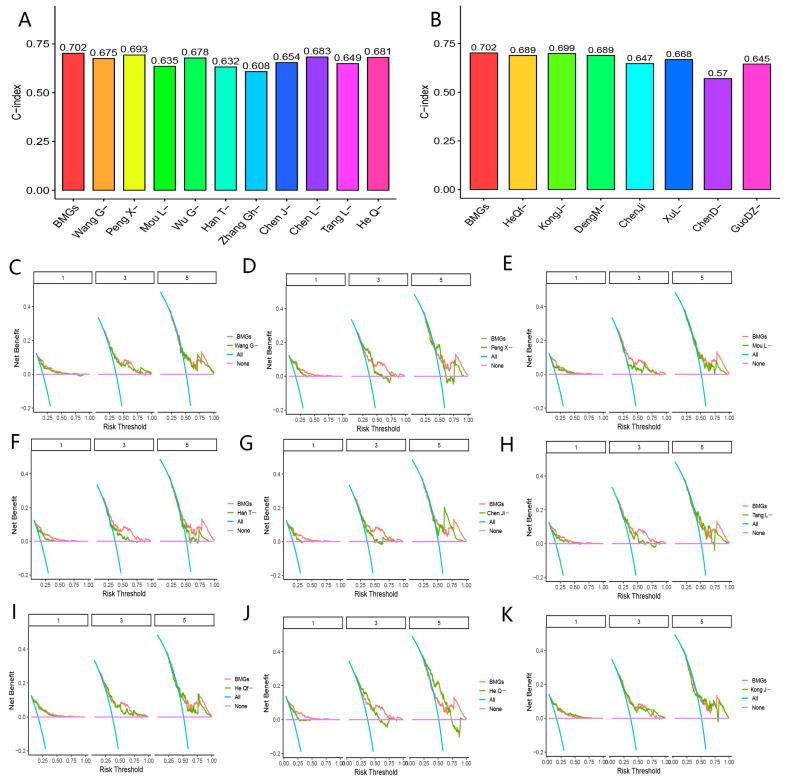
Comparison with other gene signatures. (**A**) C-index analyses about BMGs and 10 gene signatures in PubMed databases for prediction of the prognoses of HCC. (**B**) C-index analyses about BMGs and seven gene signatures in PubMed databases for prediction of the prognoses of HCC. (**C**) Decision curve analyses of BMGs and the Wang G-study. (**D**) Decision curve analyses of BMGs and the Peng X-study. (**E**) Decision curve analyses of BMGs and the Mou L-study. (**F**) Decision curve analyses of BMGs and the Han T-study. (**G**) Decision curve analyses of BMGs and the Chen Ji-study. (**H**) Decision curve analyses of BMGs and the Tang L-study. (**I**) Decision curve analyses of BMGs and the He Qf-study. (**J**) Decision curve analyses of BMGs and the He Q-study. (**K**) Decision curve analyses of BMGs and the Kong J-study.

**Figure 11 biomolecules-13-00052-f011:**
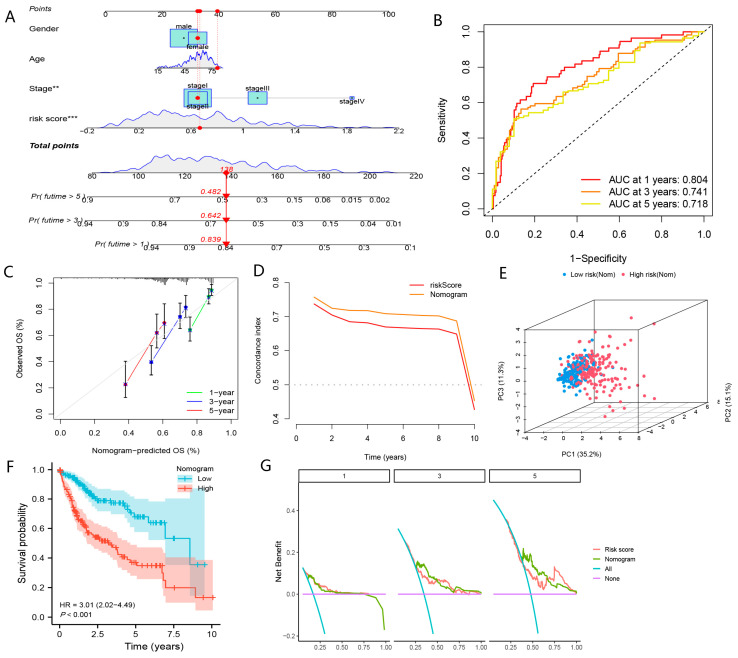
Construction of nomogram based on BMGs and other clinical features. (**A**) Construction of nomogram based on risk scores, TNM-stage, gender, and age. (**B**) The timeROC curve of nomogram. (**C**) The calibration curve of nomogram. (**D**) The c-index curve of nomogram. (**E**) PCA plot about nomogram. (**F**) KM curve of OS in high-risk (Nomogram) and low-risk (Nomogram) group. (**G**) DCA of nomogram for prediction of prognoses. (“**”, “***” represent *p* < 0.01, *p* < 0.001, respectively).

**Figure 12 biomolecules-13-00052-f012:**
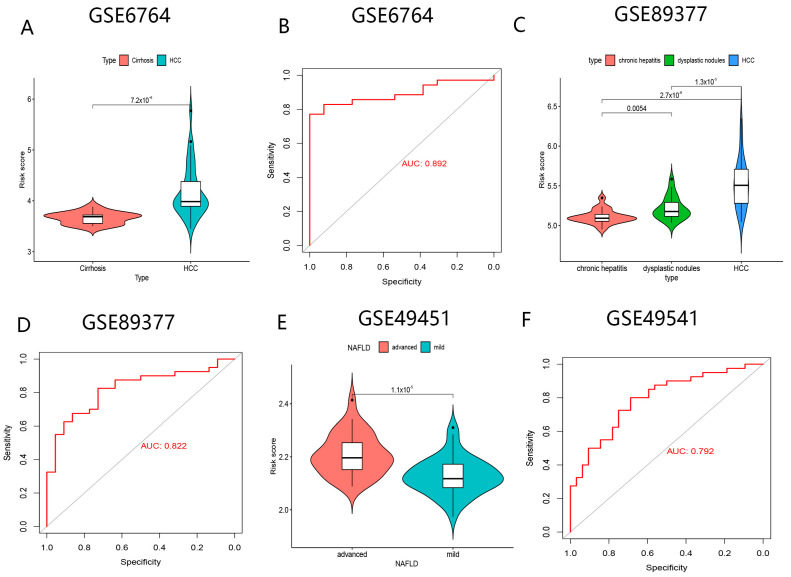
The power of BMGs for early detection the risk of HCC. (**A**) Comparison of risk scores between cirrhosis and HCC. (**B**) The ROC curve of risk score distinguishing from HCC and cirrhosis. (**C**) Comparison of risk score in cirrhosis, dysplatic node disease, and HCC. (**D**) The ROC curve of risk score distinguishing from dysplastic nodule disease and HCC. (**E**) Comparison of risk score in early and advanced steatohepatitis. (**F**) The ROC curve of risk score distinguishing from early steatohepatitis and advanced steatohepatitis.

**Figure 13 biomolecules-13-00052-f013:**
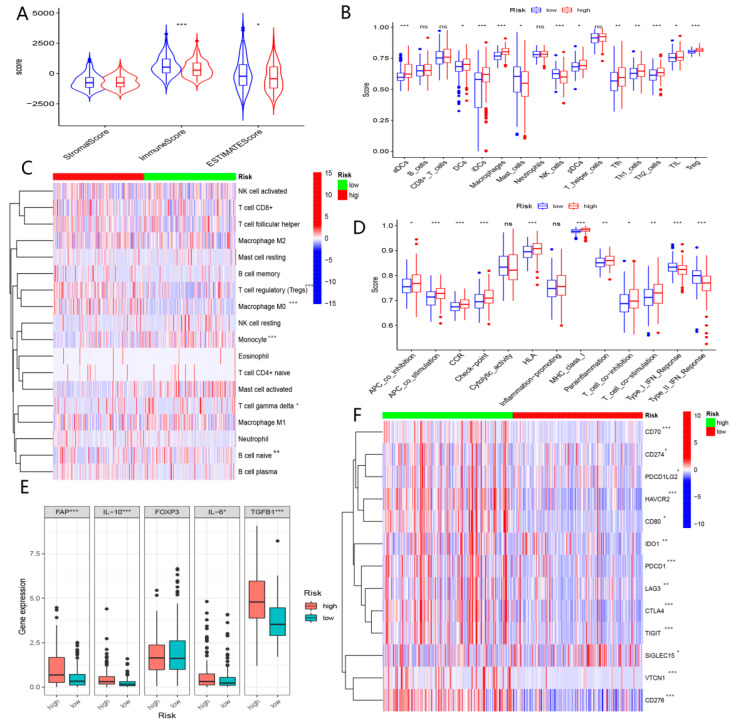
The role of BMGs in immune profiles. (**A**) Comparison of the immune microenvironment of tumors in low- and high-risk groups with the ESTIMATE algorithm. (**B**) Comparison of immune cells in low- and high-risk groups using the ssGESA algorithm. (**C**) Comparison of immune cells in low- and high-risk groups using the CIBERSORT algorithm. (**D**) Comparison of immune functions in low- and high-risk groups using the ssGESA algorithm. (**E**) Comparison of expression level of Immunosuppression-related genes in low- and high-risk groups. (**F**) Comparison of the expression level of common immune checkpoint in low- and high-risk groups. (“*”, “**”, “***” represent *p* < 0.05, *p* < 0.01, *p* < 0.001, respectively, ns: not significant).

**Figure 14 biomolecules-13-00052-f014:**
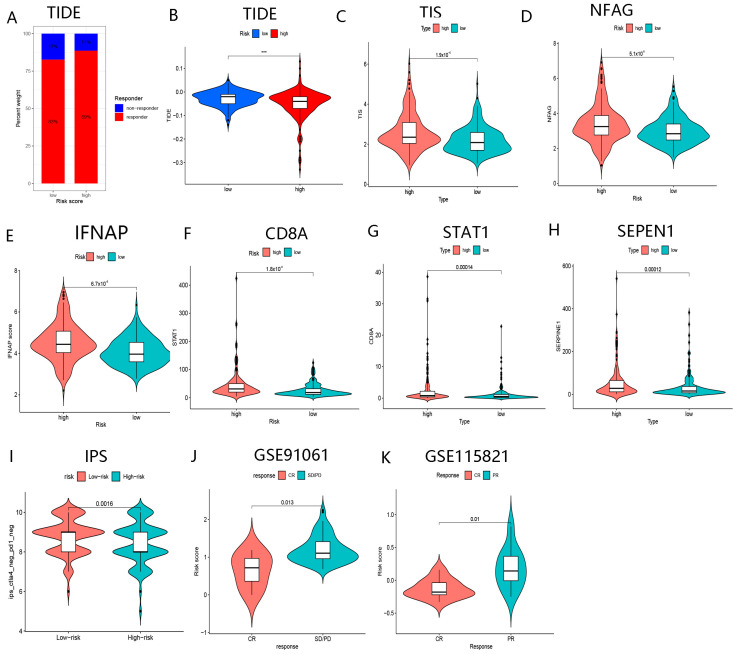
The role of BMGs for the prediction of responses to immunotherapy. (**A**) The ratio of responders and non-responders to immunotherapy in the high- and low-risk groups, respectively. (**B**) Comparison of TIDE scores in low- and high-risk groups. (**C**) Comparison of TIS scores in low- and high-risk groups. (**D**) Comparison of NFAG in low- and high-risk groups. (**E**) Comparison of IFNAP scores in low- and high-risk groups. (**F**) Comparison of the expression level of CD8A in low- and high-risk groups. (**G**) Comparison of the expression level of STAT1 in low- and high-risk groups. (**H**) Comparison of the expression level of SEPEN1 in low- and high-risk groups. (**I**) Comparison of IPS scores in low- and high-risk groups. (**J**) Comparison of risk scores in CR and SD/PD after receiving immunotherapy in the GSE91061 cohort. (**K**) Comparison of risk scores in CR and SD/PD after receiving immunotherapy in GSE115821 cohort. (“***”, represent *p* < 0.001).

**Figure 15 biomolecules-13-00052-f015:**
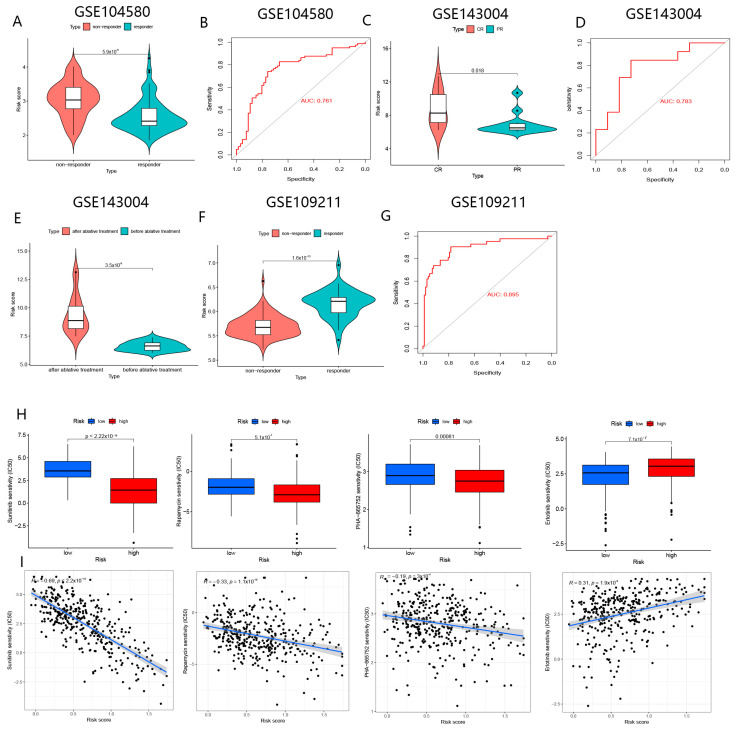
The role of BMGs for the prediction of responses to TACE and ablation therapy. (**A**) Comparison of risk scores in responders and non-responders to TACE. (**B**) The ROC of risk scores for assessing the responses to TACE. (**C**) Comparison of risk scores in CR and PR to ablation therapy. (**D**) The ROC of risk scores for assessing the responses to ablation therapy. (**E**) Comparison of risk scores before and after ablation treatment. (**F**) Comparison of risk scores in responders and non-responders to “sorafila” in GSE109211. (**G**) The ROC of risk scores for assessing the responses to “sorafila” in the GSE109211 cohort. (**H**) Comparison of IC50 of drugs in high- and low-risk groups. (**I**) The correlation of risk score with IC50 of drugs.

**Figure 16 biomolecules-13-00052-f016:**
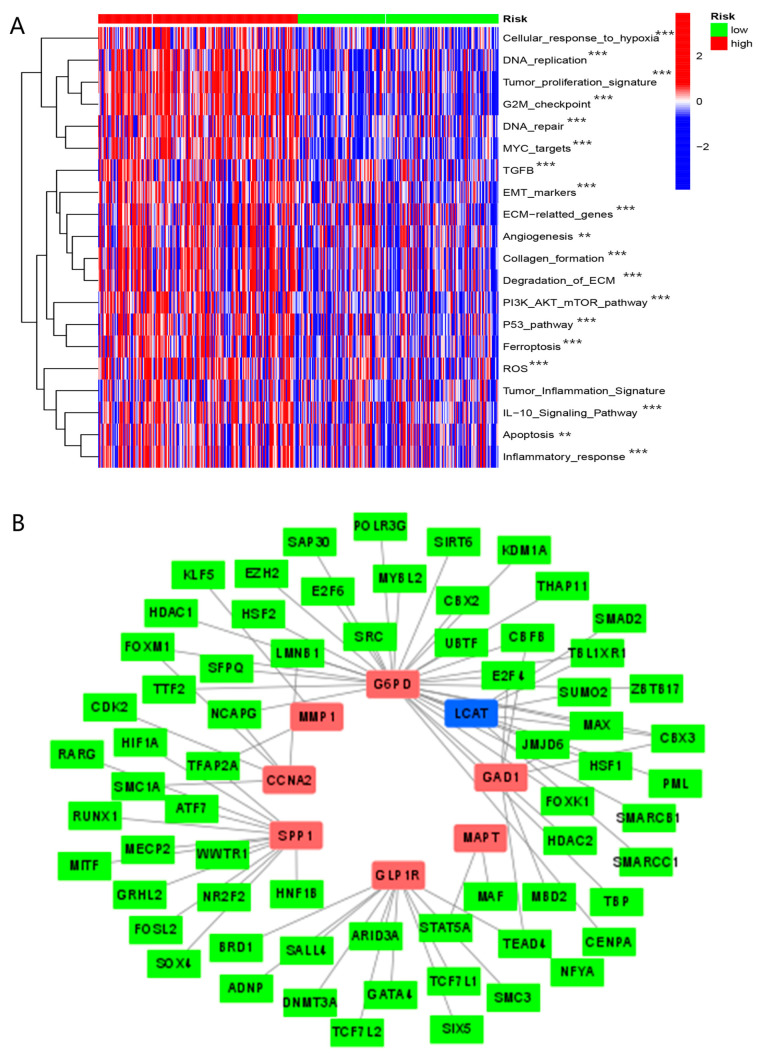
The molecular functions analyses. (**A**) Comparison of enrichment scores of tumor-associated pathways in high- and low-risk groups with the ssGSEA algorithm. (**B**) Network of genes constituting BMGs with specific transcription factors. (“**”, “***” represent *p* < 0.01, *p* < 0.001, respectively).

## Data Availability

The data used to support the findings of this study are available from the corresponding authors upon request.

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
