# Peer review of "Genes Modulating Butyrate Metabolism for Assessing Clinical Prognosis and Responses to Systematic Therapies in Hepatocellular Carcinoma"

_biomolecules, 2022, doi:10.3390/biom13010052_

Round 1

Reviewer 1 Report

In this manuscript, the authors studied the prognostic roles of butyrate metabolism (BM) related genes in hepatocellular carcinoma (HCC) using published cohorts. They identified a group of patients that overexpressed butyrate-related genes, and this group showed poor response to chemotherapy and may benefit from immunotherapy. Overall, the analyses are solid.

Below are specific comments: 

  1. It would be interesting to compare the BM-based classification to previous molecular subclasses (e.g. three integrated iClusters in PMID: 28622513 and Hoshida subtypes (PMID: 19723656)). 

  2. The description of the bioinformatic analyses should be improved. For instance, how did they perform the CIBERSORT analysis to estimate the abundance of immune subtypes, what were the inputs, did they create a reference signature matrix or used a pre-built one? The same applies to other analyses.

  3. The manuscript needs extensive revision for language.

Author Response

Dear Dr. Zhao and reviewers,
Thanks very much for your attention and the reviewers’ comments on our paper submitted to biomolecules (manuscript ID: biomolecules-2082425). We have revised the manuscript according to your kind advice and the reviewers’ comments. We have also proof-read the manuscript carefully to minimize grammatical errors. All the authors have read and approved the revisedversion of manuscript, and agreed to the authorship. We sincerely hope this manuscript will be finally acceptable to be published on biomolecules. Thanks very much for all your help and looking forward to hearing from you soon.
Sincerely,
Jing Tao, M.D., Ph.D.
Associate Professor
Department of Pancreatic Surgery, Renmin Hospital, Wuhan University
Here below is our description on revision according to the reviewers’comments.

Reviewer #1: In this manuscript, the authors studied the prognostic roles of butyrate metabolism (BM) related genes in hepatocellular carcinoma (HCC) using published cohorts. They identified a group of patients that overexpressed butyrate-related genes, and this group showed poor response to chemotherapy and may benefit from immunotherapy. Overall, the analyses are solid.
1.It would be interesting to compare the BM-based classification to previous molecular subclasses (e.g. three integrated iClusters in PMID: 28622513 and Hoshida subtypes (PMID: 19723656)). 
Response: Thanks for your helpful suggestion. We included BM-Clusters, iClusters and Hoshida-clusters in the univariate cox regression analysis, and we discovered that only BM subtype was a risk factor for prognoses of hepatocellular carcinoma (HCC) (detailed in supplementary Figure S1A). And Clinical decision analysis showed that BM subtypes have better clinical value than iCluster and Hoshida in predicting the prognosis of HCC (detailed in supplementary Figure S1B). This also illustrates the greater performances of BM-cluster for assessing the prognosis of HCC, compared to iCluster and Hoshida-cluster.
2.The description of the bioinformatic analyses should be improved. For instance, how did they perform the CIBERSORT analysis to estimate the abundance of immune subtypes, what were the inputs, did they create a reference signature matrix or used a pre-built one? The same applies to other analyses.
Response: Thanks for your helpful suggestion. We have carefully checked the manuscript and found that a small number of statistical analysis methods were indeed not described clearly in this study. We have revised the issues you pointed out. For example, we have illustrated the reference gene set for the CIBERSORT algorithm other algorithms in the methodology.
3.The manuscript needs extensive revision for language. 
Response: Thanks for your helpful suggestion. We have invited to native English speaker to check the manuscript throughout carefully again, and revised the grammar, equation, and typesetting errors, etc. And our team have also proof-read the manuscript carefully to minimize grammatical errors.

Reviewer 2 Report

This manuscript aimed to identify butyrate metabolism related genes and evaluate the prediction and prognosis of these genes in HCC. Authors sorted out BM-related genes, then narrowed down to 8 BMGs, and assessed the prognoses of 8 BMGs in TCGA and other cohorts. Also, authors comprised more analysis on immunotherapy, TACE, ablation therapy and chemotherapy which makes it more informative and comprehensive.

However, there are several problems.

1.     It is confusing which cohorts were testing datasets and which cohorts were validation cohorts. Authors used different cohorts for different validations, for example, ICGC, GSE25097, GSE87630 and CCLE were used for the validation of BMGs. But for BMGs risk score, why used TCGA, ICGC and GSE9843? Simply showing some potentially cherry-picked results is not informative. This type of study shall show overall analysis and clear patterns across multiply cohorts. Were DEGs of 757 genes identified from the comparison of tumor vs. normal, but 41 candidate genes were screened in tumor samples only? A schematic overview for comprehensive identification, validation and prediction should be added.

2.     Were there any significant improvements of prognoses using BMGs/BMGs-based nomogram compared to existing signatures in both testing and validation datasets?

3.     No computer codes were provided. It is crucial to deposit well-documented codes to a public repository for this type of study.

Some other concerns:

1.     No explanation of CAF even when it was mentioned first time.

2.     What are Li-7, JHH-5 et al?

3.     Was GSE10141 used for any validation?

4.     Didn’t explain Figure 6J in main text or in the legend. And Figures 6J-K should be Figures 6K-L.

5.     Missing Figure 9A in Figure 9’s title.

Author Response

Dear Dr. Zhao and reviewers,
Thanks very much for your attention and the reviewers’ comments on our paper submitted to biomolecules (manuscript ID: biomolecules-2082425). We have revised the manuscript according to your kind advice and the reviewers’ comments. We have also proof-read the manuscript carefully to minimize grammatical errors. All the authors have read and approved the revised
version of manuscript, and agreed to the authorship. We sincerely hope this manuscript will be finally acceptable to be published on biomolecules. Thanks very much for all your help and looking forward to hearing from you soon.
Sincerely,
Jing Tao, M.D., Ph.D.
Associate Professor
Department of Pancreatic Surgery, Renmin Hospital, Wuhan University
Here below is our description on revision according to the reviewers’comments.

Reviewer #2: This manuscript aimed to identify butyrate metabolism related genes and evaluate the prediction and prognosis of these genes in HCC. Authors sorted out BM-related genes, then narrowed down to 8 BMGs, and assessed the prognoses of 8 BMGs in TCGA and other cohorts. Also, authors comprised more analysis on immunotherapy, TACE, ablation therapy and chemotherapy which makes it more informative and comprehensive. However, there are several problems.
1.It is confusing which cohorts were testing datasets and which cohorts were validation cohorts. Authors used different cohorts for different validations, for example, ICGC, GSE25097, GSE87630 and CCLE were used for the validation of BMGs. But for BMGs risk score, why used TCGA, ICGC and GSE9843? Simply showing some potentially cherry-picked results is not informative. This type of study shall show overall analysis and clear patterns across multiply cohorts. Were DEGs of 757 genes identified from the comparison of tumor vs. normal, but 41 candidate genes were screened in tumor samples only? A schematic overview for comprehensive identification, validation and prediction should be added. 
Response: Thanks for your helpful suggestion. We carefully checked the datasets used in this study. In this study, the TCGA-LIHC cohort is the training set cohort, while the others are the validation set cohorts. We have added the necessary content in the Methodology, and Figure 1. And the schematic overview for comprehensive identification, validation and prediction was shown in Figure 1. 

In addition, the main principles of the statistical analysis of this study were to maximize the use of clinical information contained in each dataset and to corroborate each dataset with each other, thus making the conclusions more rigorous. Multiple datasets on hepatocellular carcinoma (HCC) were included in this study. Since the clinical information contained in each dataset was different, we performed targeted statistical analyses based on the clinical information contained in each dataset. For example, the GSE87630 cohort contains only the gene expression information of HCC tissues and tumor tissues, and contains no other clinical information. We can only apply it to verify whether the eight genes that constitute BMGs are differentially expressed in HCC tissues and normal tissues, and thus cannot use this dataset to verify the BMGs model to predict prognoses of HCC.
Moreover, we presumed that genes that are significantly higher or lower expressed in HCC tissues compared to normal liver tissues might be potential genes that affect the prognoses of HCC, so we first obtained 75 differentially expressed genes (DEGs) from 757 genes that regulate butyric acid metabolism. To explore which of these 75 DEGs were associated with prognoses of HCC, we included these 75 DEGs in univariate Cox regression analysis to obtain 41 candidate genes that were associated with prognoses of HCC (P<0.05). We searched for similar studies [1-4] and discovered that these studies also used the same approach to screen for genes associated with prognoses.
2.Were there any significant improvements of prognoses using BMGs/BMGs-based nomogram compared to existing signatures in both testing and validation datasets?
Response: In our study, based on the TCGA-LIHC cohort (testing dataset), BMGs/BMGs-based nomogram is demonstrated to be superior to other existed signatures in predicting prognoses of HCC. There are significant improvements of predicting prognoses fo HCC using BMGs/BMGs-nomogram compared to existing signatures in testing data set. And the superior predictive performances of BMGs to other 33 gene signatures have not been verified in the validation sets, which is also a limitation of our study. And we have discussed the limitation in the “Discussion” section. 
Notably, we searched many published studies in the PubMed database, including 33 studies included in our study, and we have discovered that these published studies did not perform similar analyses. In detail, these published studies did not compare created gene signatures with the existing signatures. Our study compared BMGs with 33 published studies to demonstrate the greater potential of BMGs to predict the prognoses of HCC, which is one of the main innovations of our study.
3.No computer codes were provided. It is crucial to deposit well-documented codes to a public repository for this type of study.
Response: Thanks for your helpful suggestion. Since the statistical analyses of this study were shared by several authors, the code was not systematically organized before that. The code is currently being sorted out by our group and will be available upon request from the corresponding author after our article is online.
Some other concerns:
1.No explanation of CAF even when it was mentioned first time.
Response: Thanks for your helpful suggestion. We have carefully checked the manuscript, and added the explanation of CAF in Page 4.
2.What are Li-7, JHH-5 et al?
Response: Thanks for your helpful suggestion. We have searched the data carefully and confirmed that Li-7 and JHH-5 are the name of the hepatocellular carcinoma cell lines.
3.Was GSE10141 used for any validation?
Response: Thanks for your helpful suggestion. According to your suggestion, we have carefully checked the manuscript and found GSE10141 to be redundant, so we have removed it.
4.Didn’t explain Figure 6J in main text or in the legend. And Figures 6J-K should be Figures 6K-L.
Response: Thanks for your helpful suggestion. We have explained Figure 6J in the main text and figure legends, and have changed Figure 6J-K to figure6K-L.
5.Missing Figure 9A in Figure 9’s title.
Response: Thanks for your helpful suggestion. We have added Figure 9A to Figue 9's title.
Reference
1. Liu Q, Yu M, Zhang T. Construction of Oxidative Stress-Related Genes Risk Model Predicts the Prognosis of Uterine Corpus Endometrial Cancer Patients. Cancers (Basel). 2022 Nov 14;14(22):5572. doi: 10.3390/cancers14225572. 
2. Zhang J, Huang C, Yang R, Wang X, Fang B, Mi J, et al. Identification of Immune-Related Subtypes and Construction of a Novel Prognostic Model for Bladder Urothelial Cancer. Biomolecules. 2022 Nov 11;12(11):1670. doi: 10.3390/biom12111670. 
3. Hu Y, Cai J, Ye M, Mou Q, Zhao B, Sun Q, et al. Development and validation of immunogenic cell death-related signature for predicting the prognosis and immune landscape of uveal melanoma. Front Immunol. 2022 Nov 16;13:1037128. doi: 10.3389/fimmu.2022.1037128. PMID: 36466923; PMCID: PMC9709208.
4. Hu X, Guo L, Liu G, Dai Z, Wang L, Zhang J, et al. Novel cellular senescence-related risk model identified as the prognostic biomarkers for lung squamous cell carcinoma. Front Oncol. 2022 Nov 17;12:997702. doi: 10.3389/fonc.2022.997702. PMID: 36465363; PMCID: PMC9712184.

Round 2

Reviewer 1 Report

The authors have made an effort to address the review comments and the paper is now much improved.

Reviewer 2 Report

My concerns were addressed. Thanks.